# Identification of *Staphylococcus aureus* Penicillin Binding Protein 4 (PBP4) Inhibitors

**DOI:** 10.3390/antibiotics11101351

**Published:** 2022-10-04

**Authors:** Mikaeel Young, Danica J. Walsh, Elysia Masters, Vijay Singh Gondil, Emily Laskey, Michael Klaczko, Hani Awad, James McGrath, Edward M. Schwarz, Christian Melander, Paul M. Dunman

**Affiliations:** 1Department of Microbiology and Immunology, University of Rochester Medical Center, Rochester, NY 14642, USA; 2Department of Chemistry and Biochemistry, University of Notre Dame, Notre Dame, IN 46556, USA; 3Center for Musculoskeletal Research, University of Rochester Medical Center, Rochester, NY 14642, USA; 4Department of Biomedical Engineering, University of Rochester, Rochester, NY 14642, USA

**Keywords:** *Staphylococcus aureus*, osteomyelitis, PBP4, antibiotic resistance

## Abstract

Methicillin-resistant *Staphylococcus aureus* (MRSA) is a global healthcare concern. Such resistance has historically been attributed to the acquisition of *mecA* (or *mecC*), which encodes an alternative penicillin binding protein, PBP2a, with low β-lactam affinity. However, recent studies have indicated that penicillin binding protein 4 (PBP4) is also a critical determinant of *S. aureus* methicillin resistance, particularly among community-acquired MRSA strains. Thus, PBP4 has been considered an intriguing therapeutic target as corresponding inhibitors may restore MRSA β-lactam susceptibility. In addition to its role in antibiotic resistance, PBP4 has also recently been shown to be required for *S. aureus* cortical bone osteocyte lacuno-canalicular network (OLCN) invasion and colonization, providing the organism with a niche for re-occurring bone infection. From these perspectives, the development of PBP4 inhibitors may have tremendous impact as agents that both reverse methicillin resistance and inhibit the organism’s ability to cause chronic osteomyelitis. Accordingly, using a whole-cell high-throughput screen of a 30,000-member small molecule chemical library and secondary assays we identified putative *S. aureus* PBP4 inhibitors. Quantitative reverse transcriptase mediated PCR and PBP4 binding assays revealed that hits could be further distinguished as compounds that reduce PBP4 expression versus compounds that are likely to affect the protein’s function. We also showed that 6.25 µM (2.5 µg/mL) of the lead candidate, 9314848, reverses the organism’s PBP4-dependent MRSA phenotype and inhibits its ability to traverse Microfluidic-Silicon Membrane-Canalicular Arrays (µSiM-CA) that model the OLCN orifice. Collectively, these molecules may represent promising potential as PBP4-inhibitors that can be further developed as adjuvants for the treatment of MRSA infections and/or osteomyelitis prophylactics.

## 1. Introduction

*Staphylococcus aureus* is a major human pathogen that is a frequent cause of community- and hospital-associated infections that result in high morbidity and mortality rates [1]. For instance, *S. aureus* bacteremia is estimated to occur within 4.3 and 38.2 per 100,000 person-years and results in 30-day mortality rates of 32–39% [2]. The organism is also a predominant cause of infective endocarditis and pneumonia, both of which result in mortality rates approaching 40% [3]. *S. aureus* is also a common cause of keratitis and osteomyelitis, which are typically not life-threatening infections but are associated with unacceptably high rates of blindness and amputation, respectively [4,5,6].

Traditionally, the β-lactam class of antibiotics has been a preferred choice of treatment for *S. aureus* infections due, in part, to their high potency and strong safety record in comparison to other antibiotics. β-lactams act as substrate analogs that covalently bind the active site- and inactivate- the transpeptidase activity of penicillin binding proteins (PBPs) [7,8]. In *S. aureus* these include four enzymes (PBP1, PBP2, PBP3 and PBP4) that catalyze peptidoglycan side-chain crosslinking, thereby conferring cell wall rigidity needed for the organism’s viability [9]. Unfortunately, the emergence and dissemination of methicillin-resistant *S. aureus* (MRSA) that have acquired an additional penicillin binding protein, PBP2a (also known as PBP2′), has virtually eliminated the therapeutic utility of β-lactam antibiotics [10,11,12]. PBP2a is a transpeptidase that, in comparison to PBP2, contains a structural rearrangement that narrows the protein’s substrate binding cleft, which in turn lowers its β-lactam affinity and allows the enzyme to maintain transpeptidase function in the presence of β-lactams [13,14].

Two fifth-generation cephalosporins, ceftobiprole and ceftaroline, have recently been developed as anti-MRSA agents with PBP2a inhibitory activity. Ceftobiprole’s PBP2a inhibitor activity is mediated by the molecule’s vinylpyrrolidinone side chain, which facilitates binding to the narrow PBP2a substrate binding cleft while positioning the lactam moiety within the active site to covalently inactivate the protein [15]. In contrast, ceftaroline allosterically induces a PBP2a conformational change that subsequently allows a second molecule to bind to the transpeptidase active site and inhibit the enzyme [16]. However, MRSA strains with intermediate and high-level ceftobiprole and/or ceftaroline resistance have begun to emerge [17,18]. Corresponding studies designed to determine the molecular mechanism(s) of resistance led to the surprising discovery of an uncanonical mode of high-level *S. aureus* β-lactam resistance (MRSA) that does not include PBP2a.

More directly, through the generation of laboratory derived high-level ceftobiprole-resistant mutants within *S. aureus* strain backgrounds lacking the gene encoding PBP2a (*mecA*), Chambers and colleagues provided one of the first indications that the organism encodes PBP2a-independent β-lactam resistance mechanisms [19]. Follow on studies revealed that such resistance was attributable to mutations leading to the overexpression of PBP4 variants harboring amino acid substitutions within the protein’s active site. In addition to ceftobiprole, these evolved strains also displayed high-level resistance to a variety of additional β-lactams as would be expected of MRSA [20,21,22,23,24]. Similarly, in a series of genetic and chemical genomic studies, Cheung and colleagues found that PBP4, as opposed to PBP2a, is required for the MRSA phenotype of community-acquired methicillin-resistant strains [25]. Moreover, small molecule inhibitors of PBP4 function and/or cellular localization were found to restore β-lactam susceptibility toward otherwise MRSA clinical isolates [25,26].

Emerging studies have revealed that in addition to modulating *S. aureus* β-lactam resistance, PBP4 may act as a virulence factor that contributes to the organism’s propensity to cause reoccurring osteomyelitis. More directly, transmission electron microscopy of animal- and clinical cases- of osteomyelitis revealed that *S. aureus* has the ability to invade cortical bone osteocyte lacuno-canalicular networks (OLCN) [27,28]. Such colonization is hypothesized to serve as a reservoir for re-occurring infection [29,30], and requires modulation of the organism’s cell wall components in a manner that allows otherwise 1 µm staphylococci to deform to approximately half their wild type size to successfully invade and colonize the narrow confines of 100–600 nm diameter canaliculi [5,31]. In a genetic screen using Microfluidic–Silicon Membrane–Canalicular Arrays (µSiM-CA) with engineered 500 nm size pores designed to mimic the OLCN orifice, Masters and colleagues found that *S. aureus* PBP4 is required for traversing the membrane pores, indicating the protein is required for OLCN colonization [32]. Indeed, mutation of *S. aureus pbp4* eliminated the organism’s ability to invade and colonize OLCN in animals [32]. As such, PBP4 has been hypothesized to represent an attractive therapeutic target for the prevention of reoccurring bone infection, as putative PBP4 inhibitors would be expected to reduce *S. aureus* OLCN invasion and, hence, eliminate a source of infection [5].

Given the importance of PBP4 in terms of both conferring antibiotic resistance and modulating osteomyelitis, we considered that small molecule inhibitors of the protein’s activities may have tremendous therapeutic value. Accordingly, herein we describe the development of a high throughput screening paradigm and secondary assays to identify members of a small molecule compound library for PBP4 inhibitors that limit both PBP4-associated antibiotic resistance and *S. aureus’* ability to traverse µSiM-CA devices, representative of the OLCN orifice.

## 2. Results

### 2.1. Screening for Small Molecule S. aureus PBP4 Inhibitors

*S. aureus* PBP4 has recently been recognized as a key mediator of β-lactam antibiotic resistance and to also be required for the organism’s ability to colonize cortical bone OLCN, a feature that has been hypothesized to provide a reservoir for reoccurring osteomyelitis. Consequently, small molecule *S. aureus* PBP4 inhibitors may represent therapeutically valuable agents that: (1) reverse antibiotic resistance, (2) eliminate chronic bone infection, and/or (3) both. Accordingly, we set out to develop a screening paradigm to identify small molecules that inhibit the enzyme’s activities.

The approach used was to initially search for compounds that reverse the antibiotic resistance profile of wild type cells to that of a *pbp4* mutant. To define screening conditions three strains were employed, a wild type USA300 MRSA strain, an isogenic *pbp4* deletion strain, and a complemented strain harboring a plasmid wildtype *pbp4* copy. The antibiotic susceptibility profile of each strain was measured to determine the magnitude with which PBP4 affects resistance to various antibiotics (Table 1). As expected, minimum inhibitory concentration (MIC) testing revealed that wild type USA300 displayed resistance to the β-lactam antibiotics nafcillin (8.0 µg/mL) and meropenem (4.0 µg/mL) and that resistance was reduced 2-fold for an isogenic *pbp4*-mutant strain, supporting earlier studies indicating that PBP4 impacts MRSA phenotype. While PBP4 seemingly did not affect tobramycin, minocycline, mupirocin, or colistin resistance, we serendipitously found that resistance to the fluoroquinolone antibiotics, ciprofloxacin and to a lesser extent levofloxacin, is likely to be impacted by PBP4. More specifically, wild type USA300 cells displayed a ciprofloxacin MIC of 4.0 µg/mL, whereas the MIC of the *pbp4* mutant strain was reproducibly determined to be 1.0 µg/mL and complementation with a wild type copy of the *pbp4* gene partially restored ciprofloxacin resistance (2.0 µg/mL) during these assay conditions. Taken together, these results indicate that PBP4 provides a 2-fold increase in resistance to nafcillin and meropenem, but a greater ~4-fold resistance against the antimicrobial effects of ciprofloxacin for the USA300 strain used here. We anticipated that the greater dynamic range of the latter could provide a means to reliably screen for agents that inhibit PBP4 function. We reasoned that compounds that have no impact on PBP4 activity would not affect USA300 growth in the presence of ciprofloxacin, whereas compounds that inhibit PBP4 function would display a loss of growth phenotype in the presence of the antibiotic, phenocopying the *pbp4*-mutant.

To explore this possibility further, Z’-factor testing was performed to determine whether an appropriate ciprofloxacin concentration could be identified that reproducibly allowed growth of wild type (PBP4+) cells, but inhibited growth of PBP4-cells, in a high throughput setting. Growth measures were performed for wild type and Δ*pbp4* mutant cells in media alone or media supplemented with 1.0, 2.0, 3.0 or 4.0 µg/mL ciprofloxacin. Media containing 2 µg/mL ciprofloxacin provided the most reliable culture conditions to distinguish between wildtype cells and the Δ*pbp4* strain, generating a Z’-factor score of 0.31 (Figure 1A). Members of a 30,000 compound ChemBridge small molecule diversity set were individually screened at 50 µM to identify molecules that reduced wild type USA300 growth to that of the Δ*pbp4* strain in media supplemented with 2.0 µg/mL ciprofloxacin. The results reveal that 29,679 of these molecules (98.9%) did not impact the organism’s growth in the presence of ciprofloxacin, whereas 321 compounds (1.1%) eliminated growth. We considered two ways in which these 321 compounds were inhibiting growth may be due to either: i. directly inhibiting PBP4 function, or ii. compounds with standalone antimicrobial activity. To enrich for PBP4 inhibitors, the standalone antimicrobial activity of each of the 321 compounds was directly evaluated in the absence of antibiotic. A total of 160 compounds (49.8%) inhibited USA300 growth at 50 µM and, because *S. aureus* PBP4 is non-essential, were removed from further consideration as PBP4-specific inhibitors. Conversely, 161 compounds (50.2%) exhibited no detectible standalone antimicrobial activity, suggesting a subset of these molecules may include PBP4-specific inhibitors.

To distinguish putative PBP4 inhibitors from non-specific antibiotic potentiators we evaluated whether the remaining 161 compounds potentiated another antibiotic, mupirocin. As shown in Table 1, there is no detectible difference in mupirocin susceptibility between wild type USA300 cells and *pbp4* mutant cells, suggesting that PBP4 inhibitors should have no impact on mupirocin susceptibility, whereas non-specific antibiotic potentiators may alter the mupirocin activity. Standard fractional inhibitory concentration (FIC) testing to measure compound interactions of all 161 compounds revealed that 65 (40.4%) potentiated the antimicrobial effects of mupirocin toward wild type USA300 cells indicating that they are non-specific antibiotic potentiators and were triaged, whereas 96 compounds (59.6%) had no impact on the strain’s susceptibility to mupirocin. Next, an early test of potential eukaryotic cell cytotoxicity was performed by measuring the effects of each of the 96 compounds for growth inhibition against yeast cells. Two compounds (~2%) inhibited *Saccharomyces cerevisiae* growth, and thus were considered likely to also be toxic toward mammalian cells and triaged, whereas the remaining 94 compounds (~98%) had no effect. Taken together, we considered that these 94 molecules represent a compound set that is likely to be enriched for *S. aureus* PBP4 inhibitors worthy of follow-on characterization.

### 2.2. Potentiation of β-Lactams

To further evaluate their potential PBP4 inhibitor activity, prioritized compounds were tested for their ability to reverse PBP4 mediated antibiotic resistance in another *S. aureus* background using three isogenic strains that became available during our study (generous gift from Dr. S. Chatterjee; University of Maryland). These strains included: (1) COLnex, which is a methicillin susceptible strain, (2) CRB, a methicillin/ceftobiprole-resistant strain of COLnex that overproduces PBP4, and (3) CRB∆*pbp4*, a methicillin susceptible strain of CRB lacking the *pbp4* gene [20,33]. As expected, MIC testing (Table 1) confirmed the resistance profiles of each of these strains to ceftobiprole as being 0.25 µg/mL (COLnex), 128 µg/mL (CRB; PBP4-overproducer) and 0.25 µg/mL (CRB∆*pbp4*), as previously reported. We reasoned that the observed 512-fold difference in ceftobiprole resistance between the PBP4-overexpressor and corresponding ∆*pbp4* derivative provided an excellent dynamic range to further evaluate whether the putative inhibitors do (or do not) display the ability to reverse PBP4-associated β-lactam resistance in a second strain background. Of note, while PBP4 seemed to modulate fluoroquinolone resistance in the USA300 strain used here (above), the presence or absence of the *pbp4* gene did not appear to significantly impact fluoroquinolone resistance in the COLnex strain set (Table 1).

Standard fractional inhibitory concentration testing was performed using *S. aureus* strain CRB in checkerboard format in which each row contained increasing concentrations of ceftobiprole (0, 1.0, 2.0, 4.0, 8.0, 16, 32, 64, or 128 µg/mL) and each column contained increasing concentrations (0, 3.125, 6.25, 12.5, 25, 50, or 100 µM) of a putative PBP4 inhibitor. Of the 94 compounds evaluated, 88 (93.6%) either did not, or only marginally reduced, CRB ceftobiprole susceptibility (2-fold MIC reduction at 100 µM putative PBP4 inhibitor), suggesting that they are not able to inhibit the strain’s PBP4 or are low potency inhibitors. However, six compounds exhibited either additive or synergistic antimicrobial activity (FICI ≤ 1) in combination with ceftobiprole, suggesting that they may be effective PBP4 inhibitors (Table 2). More specifically, while CRB growth was not affected by 128 µg mL^−1^ ceftobiprole (alone) or ≥100 µM each compound (alone), the strain’s ceftobiprole MIC decreased from 128 to 4.0–64 µg/mL when combined with 12.5 to 50 µM each putative PBP4 inhibitor. Compound 5784306 reduced ceftobiprole resistance 2-fold, compound 7611906 reduced resistance 8-fold, compounds 9009498, 7974147, and 9314848 reduced resistance 16-fold, and compound 5784306 reduced resistance 32-fold (all compound numbers refer to ChemBridge chemical identifiers). To establish whether these compound-associated reductions in strain CRB ceftobiprole resistance were PBP4 dependent, FIC testing was repeated using strain CRB∆*pbp4* (Table 2). Four compounds had no impact on ceftobiprole activity toward the deletion strain, whereas one compound provided a modest 2-fold decrease in resistance, suggesting their ability to reduce CRB ceftobiprole resistance was primarily due to PBP4 inhibition. Similarly, while compound 9009498 decreased ceftobiprole resistance of the *pbp4* mutant strain 4-fold, the compound reduced resistance of wild type USA300 16-fold, indicating that the compound affects PBP4-mediated resistance. Thus, all compounds were carried forward for further characterization.

### 2.3. Human Cell Cytotoxicity Measures of Putative PBP4 Inhibitors

Although none of the six putative PBP4 inhibitors overtly affected the growth of yeast cells, we set out to more formally evaluate whether they display cytotoxicity toward human cells as a means to further narrow the list of compounds of interest. To do so, standard MTT assays were performed using human HepG2 hepatocellular carcinoma cells in the presence of 0, 50, 100, 200, and 400 µM of each compound. Following International Organization for Standardization guidelines, compounds that resulted in cell viability below 70% were considered to exhibit human cell cytotoxicity [34]. As shown in Figure 1B, none of the six compounds were toxic toward HepG2 cells at ≤100 µM, which is at least 2× concentration required to reduce strain CRB PBP4-associated ceftobiprole resistance. Thus, we considered that all six compounds were worthy of follow on study.

### 2.4. Effects of Putative PBP4 Inhibitors on pbp4 Expression

Given that each of the six compounds appeared to affect PBP4-mediated antibiotic resistance, we considered that their effects may be modulated by either inhibiting the protein’s function or by reducing *pbp4* expression, the latter of which may be readily overcome by regulatory mutations that could in-turn easily result in resistance. Thus, we sought to prioritize compounds that may affect the protein’s function and de-prioritize compounds that affect the protein’s expression. Quantitative real-time reverse transcriptase PCR (qRT-PCR) was used to measure the *pbp4* transcript titers within USA300 and CRB cells following compound treatment (Figure 2). The results indicate that treatment with three compounds reduced *pbp4* expression in both CRB and USA300, whereas one compound (77295377) slightly reduced USA300 *pbp4* expression but significantly reduced expression in CRB. However, two compounds, 7974147 and 9314848, did not seem to affect *pbp4* expression, suggesting that they may directly affect the protein’s function. We chose to focus the remainder of our studies on 9314848 (Figure 3A) because the molecule appeared the more druggable and chemically tractable for downstream structure activity relationship studies.

### 2.5. Effect of PBP4 Inhibitors on Triton X-100 Susceptibility

Cheung and colleagues have shown that *S. aureus* tolerance to Triton X-100 is mediated by PBP4 [25]. More directly, wild type cells are tolerant of 0.1% of the detergent, while *pbp4* mutant cells are not. Accordingly, as a first test of whether 9314848 may affect PBP4’s cellular function, we evaluated whether compound treated wild type cells phenocopy the growth defect of *pbp4* mutant cells in the presence of 0.1% triton X-100 (Figure 3B). As expected, wild type USA300 displayed a robust growth phenotype in the absence or presence of the detergent, whereas *pbp4* mutant cells displayed wild type growth in the absence of the detergent but a growth defect in media supplemented with 0.1% Triton X-100. Similarly, 9314848 treated wildtype cells did not display a growth defect in the absence of detergent but exhibited growth defect in the presence of Triton X-100, recapitulating the growth characteristics of *pbp4* mutant cells in the presence of detergent.

### 2.6. Ability of PBP4 Inhibitors to Bind to PBP4 or PBP2A

To more directly evaluate whether 9314848 is likely to affect PBP4 function, the compound’s ability to bind recombinant PBP4 was measured using BOCILLIN FL fluorescence polarization displacement assays [35]. The premise for the assay is that the fluorescence polarization of the fluorescently labeled β-lactam, penicillin V (BOCILLIN FL), will increase upon binding to its cognate target (i.e., PBP4). Further, compounds that bind PBP4 and inhibit its function would impede BOCILLIN FL binding thereby allowing us to assess the protein binding affinity of 9314848 using a simple fluorescence reduction assay.

As shown in Figure 4A, control reactions showed that BOCILLIN FL (alone) exhibited relatively low fluorescence polarization (~100 relative fluorescent units; RFUs), but increased 3-fold following incubation with recombinant *Bacillus subtilis* PBP4 (310 RFU), as expected. To determine whether the assay is amenable to measuring the binding properties of a compound that is known to bind PBP4 vs. a compound that is not expected to bind the protein we performed fluorescence polarization assays in which PBP4 was preincubated with either the β-lactam ampicillin, or the aminoglycoside kanamycin. As expected, preincubation with ampicillin, which is known to covalently bind the PBP4 active site, limited BOCILLIN FL binding. Conversely, preincubation with kanamycin (which does not bind PBP4) had no impact on BOCILLIN FL binding to PBP4, resulting in a high fluorescent polarization signal. Tests of 9314848 in which the compound was preincubated with PBP4 followed by the addition of BOCILLIN FL revealed that the compound limited fluorescence polarization, indicating that 9314848 either is a competitive inhibitor- or binds in an allosteric site that alters PBP4 architecture in a manner- that inhibits BOCILLIN FL binding.

A dose response study was conducted (Figure 4B) in which PBP4 was preincubated with 0, 3.125, 6.25, 12.5, 25, or 50 µM of ampicillin, kanamycin, or 9314848. As expected, ampicillin treatment exhibited a smooth PBP4 binding dose curve reaching maximum BOCILLIN FL binding inhibition at 25–50 µM, whereas kanamycin had no inhibitory effect on PBP4 BOCILLIN FL binding. Minimal binding of 9314848 to PBP4 was observed at 3.125 µM and 6.25 µM; however increased binding was observed at all concentrations above 6.25 µM. At 50 µM, 9314848 reduced BOCILLIN FL binding by ~66% and the IC_50_ was calculated to be 13.4 µM. As a control, we evaluated the ability of 50 µM 9314848 to bind to PBP2A and found that the compound displayed low and inconsistent inhibition of BOCILLIN FL labeling of PBP2A (25.81 ± 11.41%).

A second, yet lower resolution, approach was used to verify the impact of 9314848 on PBP4 substrate binding in which the protein was preincubated with compound or control antibiotics then labeled with the BOCILLIN FL and resulting fluorescent labeled protein was visualized following gel electrophoresis. As expected, in the absence of compound BOCILLIN FL readily labeled recombinant PBP4, whereas preincubation of the protein with the β-lactam antibiotics ampicillin (not shown) or cefoxitin reduced the protein’s ability to bind the fluorescent substrate in a dose-dependent manner (Figure 4C,E). Conversely, preincubation of PBP4 with 2000 µM or 5000 µM (not shown) kanamycin did not significantly impact the protein’s ability to bind BOCILLIN FL. As shown in Figure 4D,E, dose response testing in which PBP4 was preincubated with 0, 50, 500, 2000, or 5000 µM of 9314848 revealed that the compound limited PBP4 BOCILLIN FL binding by 37.50 ± 8.20% (50 µM) to 72.12 ± 13.09% (5000 µM). Of note, treatment of PBP4 with ampicillin, cefoxitin, or 9314848 at concentrations below 50 µM did not reproducibly demonstrate BOCILLIN FL binding inhibition.

### 2.7. PBP4 Inhibitors Prevent Propagation in µSiM-CA Canaliculi Model

While the above results collectively indicate that 9314848 binds PBP4 and significantly reduces PBP4-associated antibiotic resistance in two *S. aureus* strain backgrounds, we set out to evaluate if the compound could also inhibit PBP4-mediated *S. aureus* canalicular colonization. As discussed above, PBP4 has been shown to be required for the organism’s transmigration through µSiM-CA devices that mimic the canalicular network orifice. Thus, we tested 9314848 for its ability to eliminate *S. aureus* migration through µSiM-CA membrane pores. As shown in Figure 5, after inoculation of 1 × 10^7^ CFUs of wild type USA300 to the top well of µSiM-CA devices, the bacteria readily migrated through the device 500 nm pores, resulting in virtually an equivalent number of cells in the device bottom chamber. Conversely, the USA300∆*pbp4* strain was unable to efficiently transverse the membrane, verifying that PBP4 is required for *S. aureus* membrane transmigration, as previously reported [32,36]. To evaluate whether 9314848 is capable of eliminating PBP4-associated *S. aureus* µSiM-CA membrane migration, 1 × 10^7^ CFU wild type cells were incubated with 0, 3.125, 6.25, 12.5, or 25 µM of the compound for 2 h then applied to the top chamber of the device. Following incubation, cells in the top and bottom chambers were enumerated. Compound treatment did not affect the viability of cells within the top/treatment chamber. Further, while treatment with ≤3.125 µM of the molecule had no effect on *S. aureus* membrane transmigration, concentrations greater than 6.25 µM (2.5 µg/mL) completely inhibited the process, suggesting the EC_50_ is between 3.125 and 6.25 µM.

## 3. Discussion

While PBP2a has historically been considered the main driver of MRSA, the recent finding that high-level *S. aureus* β-lactam resistance occurs in strains not capable of producing PBP2a has led to investigation of non-canonical pathway(s) of MRSA development as well as fundamental questions regarding the clinical diagnostic practices that rely on detection of *mecA*, the gene encoding PBP2a, as a MRSA determinant [37]. Recent studies have revealed that development of PBP2a-independent MRSA is associated with PBP4 production in community-acquired MRSA isolates and/or overproduction of a PBP4 variant within the hospital-acquired strain COL background. With regard to the latter, the COL derivative, CRB, is a MRSA strain that lacks *mecA* but contains a 36-base pair duplication upstream of the PBP4 open reading frame and two active site substitutions (E183A and F241R) that are thought to play a central role in conferring the strain’s MRSA phenotype. The PBP4 missense mutations seem to impair binding of ceftobiprole approximately 150-fold, whereas the upstream mutation is thought to lead to an approximately 145-fold increase in PBP4 production, which in-turn elicits ceftaroline resistance [24,38]. Thus, the combined activity of both overproduction of a PBP4-derivative produces resistance to both antibiotics as well as other β-lactams within CRB. Agents that limit PBP4 activity and/or the PBP4 variant within strain CRB could reverse the PBP2a-independent pathway of MRSA development. Such agents may have tremendous therapeutic value as adjuvants dosed concurrently with β-lactams that are effective against PBP2a, such as ceftobiprole.

In addition to modulating β-lactam resistance, emerging studies indicate that PBP4 may be a previously overlooked *S. aureus* virulence factor that plays a critical role in osteomyelitis pathogenesis, a debilitating musculoskeletal infection with a prevalence of approximately 22 cases per 100,000 person-years [39]. More directly, in a ground-breaking study designed to determine the bacterial source of re-occurring osteomyelitis, it was found that *S. aureus* has the remarkable ability to invade and colonize cortical bone osteocyte lacuno-canalicular networks (OLCN), which is hypothesized to provide a reservoir for the organism to cause persistent infection [27,28]. PBP4 was subsequently determined to be essential for OLCN invasion, both in an in vitro (µSiM-CA) model system and in animals, providing a promising target for therapeutic development to combat osteomyelitis [32]. Further, cells lacking PBP4 expression have less cross-linked peptidoglycan and decreased cell wall stiffness, suggesting that cell wall metabolism plays a key role in the organism’s ability to invade sub-micron canaliculi [40]. Recently, µSiM-CA and follow-on animal studies have recently revealed that another penicillin binding protein, PBP3, which interfaces with RodA to mediate *S. aureus* side-wall peptidoglycan synthesis, is also required for OLCN invasion and colonization [36,41]. Thus, agents that inhibit PBP4 and/or PBP3 function may represent valuable therapeutics that reduce *S. aureus*’ ability to cause reoccurring osteomyelitis; however, PBP3 is not associated with high-level β-lactam resistance.

In the current study we set out to develop a screening approach to identify small molecule inhibitors of *S. aureus* PBP4, based on the premise that such agents may have dual therapeutic utility that have the ability to reverse β-lactam resistance and/or reduce osteomyelitis pathogenesis. To that end, it has been previously shown that cefoxitin, a β-lactam that binds PBP4 with high affinity, reversed the PBP4 mediated oxacillin resistance phenotype of community-acquired MRSA strains [25]. By extension, we anticipated that novel PBP4 inhibitors would similarly reverse *S. aureus* PBP4-associated antibiotic resistance. Using a MRSA USA300 strain as a model screening organism we found that indeed PBP4 modulates the organism’s β-lactam resistance phenotype, but that the protein was associated with greater resistance to the fluoroquinolone, ciprofloxacin, which was an unexpected finding that, to our knowledge, has not been previously reported. Nonetheless, others have found that PBP4 mediates *Erwinia* sp. Fluoroquinolone resistance (William Johnson, personal communication). Presumably, the previously observed decrease in peptidoglycan crosslinking of PBP4 deficient cells facilitates the antibiotic’s cellular entry, although this was not formally evaluated and was not a generalizable phenotype for other antibiotic classes, such as mupirocin, that target cytoplasmic enzymes.

A screen of 30,000 compounds identified agents that potentiated the antimicrobial activity of ciprofloxacin toward wild type USA300, and also potentiated the activity of nafcillin toward the strain but had no impact on the antimicrobial activity of mupirocin, effectively phenocopying a USA300∆*pbp4* strain. Follow-on studies performed using CRB (PBP4-variant over-expresser) determined that a subset of compounds also reduced the strain’s MRSA phenotype. While CRB was resistant to 128 µg/mL of ceftobiprole, resistance decreased to 4–16 µg/mL in the presence of 12.5–25 µM each compound. Three compounds limited *S. aureus pbp4* transcription, providing an indication that the screening paradigm effectively identifies agents that modulate PBP4 activity either directly or indirectly; identifying the cellular targets of these three compounds is expected to provide insight regarding the regulatory networks that modulate *S. aureus* PBP4 expression. Two compounds, 7974147 and 9314848 did not significantly impact *pbp4* transcription, suggesting that they may affect the protein’s function.

For the current study, we chose to focus attention on characterizing compound 9314848, which reduced strain USA300 ceftobiprole resistance from 1 µg/mL to 0.25 µg/mL and strain CRB ceftobiprole resistance from 128 µg/mL to 8 µg/mL and appeared to be *S. aureus* specific, as the compound had no effect on *Enterococcus faecalis* or *Pseudomonas aeruginosa* ceftobiprole susceptibility (not shown). As a means to further evaluate whether the compound affects PBP4 activity a series of assays were performed. First, Triton X-100 studies indicated that 9314848 increases wild type USA300 susceptibility to the detergent to a level also observed with USA300∆*pbp4* cells. Second, BOCILLIN FL fluorescence polarization studies indicated that the compound was capable of binding recombinant *B. subtilis* PBP4 protein. Third, using µSiM-CA devices, it was found that the compound limited *S. aureus* membrane transmigration. From these perspectives, 9314848 appears to limit *S. aureus* PBP4-associated resistance and migration through devices mimicking cortical bone OLCN, suggesting that the molecule may represent a promising chemical scaffold for advancing to medicinal chemistry-based optimization and refinement.

## 4. Materials and Methods

### 4.1. Bacterial Strains and Chemicals

The bacterial strains used in this study are listed in Table 3. *S. aureus* strain USA300 is a predominant cause of U.S. community-associated MRSA infections, whereas strain COL is a commonly studied hospital-associated MRSA strain. The *S. aureus* USA300 *pbp4*-null strain (USA300Δ*pbp4*) and PBP4 complementation strain (USA300∆*pbp4* pPBP4) have been previously described [32]. Strain COLn is a laboratory derived tetracycline susceptible derivative of COL, whereas COLnex is a methicillin susceptible COLn derivative that has been cured of SCCmec thereby removing the *mecA*, which encodes PBP2a, and has been previously described [33]. Strain CRB is a well-studied MRSA COLnex derivative that contains two amino acid substitutions (E183A and F241R) near the PBP4 active site and a 36-bp duplication 290-bp upstream of the *pbp4* open reading frame leading to 20–40-fold overexpression of the gene [19,20,24]. All strains were grown in Mueller Hinton (MH) broth at 37 °C, as indicated. Penicillin, streptomycin, ciprofloxacin, mupirocin, meropenem, and mitomycin C were purchased from Fisher Scientific (Waltham, MA, USA), whereas ceftobiprole was obtained from MedChemExpress LLC (Monmouth Junction, NJ, USA). The 30,000-member DIVERSET-EXP small molecule chemical library (Blocks 3–5) used in these studies was purchased from ChemBridge Corporation (San Diego, CA, USA).

### 4.2. High Throughput Screen for PBP4 Inhibitors

A 30,000-member ChemBridge small molecule compound library was screened for agents that potentiated the antimicrobial performance of ciprofloxacin toward *S. aureus* strain USA300, effectively phenocopying a USA300∆*pbp4* strain. As a prerequisite for screening, Z-factor analyses were performed to identify the optimal ciprofloxacin concentration that reproducibly allowed distinction between the growth of USA300 and USA300∆*pbp4* strains. To do so, in 96-well format strains USA300 and the *pbp4* deletion strain were used to inoculate (~1 × 10^4^ colony forming units; CFUs) wells of alternative rows of a microtiter plate containing 100 µL MH media (final volume) supplemented with either 0, 1, 2, or 4 µg mL^−1^ ciprofloxacin. Plates were incubated overnight, bacterial growth within individual wells was measured by OD_600nm_, and Z-factor was calculated; the results reveal that MH media supplemented with 2 µg mL^−1^ ciprofloxacin reproducibly distinguished between wild type and *pbp4* mutant cells (Z-factor of 0.314). To screen for compounds that phenocopied the ciprofloxacin susceptibility of *pbp4* mutant cells, a total of 1 × 10^4^ CFUs of strain USA300 (10 µL) were inoculated into individual wells of a 96-well microtiter plate containing 88 µL of MH media supplemented with 2 µg mL^−1^ ciprofloxacin, and 2 µL of the test compound was added (50 µM final concentration). Plates were incubated at 37 °C for 16 h and growth was measured by the naked eye. Compounds that prevented growth were considered hits. To counter select for compounds with inherent antimicrobial activity, hits were directly assessed for stand-alone antimicrobial activity toward *S. aureus* USA300 by repeat testing (50 µM) in the absence of ciprofloxacin.

### 4.3. Eukaryotic Toxicity Testing

Two rounds of eukaryotic toxicity testing were performed. Initially hits of interest were evaluated for antifungal activity in high throughput manner. To do so, 10^5^ CFU of *Saccharomyces cerevisiae* YSB1001 [44] cells (10 µL) were inoculated into individual wells of a microtiter plate containing yeast peptone dextrose media (88 µL) and 50 µM of each hit (2 µL). Plates were incubated at 37 °C for 16 h and growth was visually measured by the naked eye; any compound which appeared to prevent growth was considered to exhibit eukaryotic cytotoxicity and triaged. More extensive mammalian cytotoxicity testing was performed on the indicated highest priority hits of interest following International Organization for Standardization guidelines [34]. Briefly, human liver epithelial cells (HEPG2) were cultured in Dulbecco’s modified Eagle medium (DMEM; Fisher Scientific) supplemented with 10% heat inactivated fetal bovine serum (FBS; Corning Life Sciences, Corning, NY, USA) and 1% penicillin and streptomycin. Cells were incubated at 37 °C with 5% CO_2_ in Nunc tissue culture flasks (Roskilde, Denmark) until reaching 70% confluency. Cells were removed with 0.25% trypsin (Fisher Scientific), resuspended in fresh medium and used to seed approximately 2.5 × 10^5^ mL^−1^ cells into individual wells of a 96 well tissue culture microplate (Nunc, Roskilde, Denmark) containing 200 μL of fresh medium, and incubated for 24 h. The media were then removed, cells were washed with 1x phosphate buffered saline (PBS) and fresh media supplemented with 5% per volume of a compound of interest at final concentrations ranging from 0 to 400 μM were added; DMSO and 125 ug mL^−1^ mitomycin C served as negative and positive toxicity controls. Mixtures were incubated for 20–24 h at 37 °C with 5% CO_2_, the media were removed and replaced with 100 μL of fresh media, and 10 μL of 12 mM (3-(4,5-dimethylthiazol-2-yl)-2,5-diphenyltetrazolium bromide (MTT) reagent (Cyquant Cell Viability kit; Invitrogen, Carlsbad, CA, USA) was added to each well. Cells were incubated at 37 °C with 5% CO_2_ for 2–4 h, 85 μL of media was removed and the remainder was mixed with 50 µL DMSO and incubated for an additional 10 min at 37 °C in the dark. Using a SPECTRAmax5 microplate reader, cell viability was recorded as absorbance at OD_540nm_. All compounds were tested in triplicate and cell viability was expressed as a percent viability of treated cells in comparison to mock treated cells.

### 4.4. Fractional Inhibitory Concentration Index (FICI)

Fractional inhibitory testing was performed in checkerboard format to determine whether putative PBP4 inhibitors potentiated the antimicrobial effects of β-lactams toward *S. aureus* strains USA300 and CRB, as previously described [45]. To do so, in triplicate 10^5^ CFU of each strain (10 µL) was added to individual wells of a microtiter plate containing 90 µL MH media. Each row of each plate was supplemented with increasing concentrations of the test PBP4 inhibitor (0, 3.125, 6.25, 12.5, 25, 50, 100, 200 and 400 µM), whereas each column was treated with increasing concentrations (0, 0.125, 0.25, 0.5, 1.0, 2, 4, 8, 16, 32, 64 and 128 µg/mL) of antibiotic (ceftobiprole, meropenem, cefepime or oxacillin). Plates were incubated at 37 °C for 16–20 h and the lowest concentration of each test agent that inhibited bacterial growth alone or in combination was determined by the naked eye. The fractional inhibitory concentration (FIC) index was calculated using the formula: FIC of A (MIC of drug A in combination/MIC of drug A alone) + FIC B (FIC of drug B in combination/MIC of drug B alone) [46]. A synergistic interaction was defined as a FICI value of less than 0.5, an additive interaction was defined as a FICI value between 0.5 and 1.0, an indifferent interaction was defined as a FICI value between 1.0 and 4.0, and an antagonistic interaction was defined as a FICI value of more than 4.0 [46].

### 4.5. S. aureus µSiM-CA Transmigration

The effect of PBP4 inhibitors on *S. aureus* migration through 500 nm nanoporous membranes modeling the orifices of osteocyte lacuno-canalicular networks was investigated using a µSiM-CA device, as previously described [32]. Briefly, µSIM-CA systems were fabricated by SiMPore Inc. (West Henrietta, NY, USA) to include a top and a bottom chamber that were separated by 400 nm thick silicon nitride membrane containing an array of 500 nm pores, allowing for quantification of bacteria that migrate from the apical (top) well to the basal (bottom) well. To measure bacterial migration through the device membrane, *S. aureus* strains USA300 or USA300∆*pbp4* were grown overnight at 37 °C with aeration, diluted (1:100) in fresh MH media, and then grown to mid-exponential phase (OD_600nm_ = 0.2) in the presence or absence of putative PBP4 inhibitor for approximately 2 h at 37 °C shaking with aeration. A total of 100 µL of cell mixture (~1 × 10^7^ CFU) was transferred into the top reservoir of the device and incubated for 6 h at 37 °C. A total of 10 µL of media from the apical and basal reservoirs was collected, serial diluted in 0.8% NaCl, and plated on MH agar plates for bacterial enumeration.

### 4.6. Quantitative Reverse Transcriptase PCR

PBP4 transcript titers of bacterial cells exposed to putative PBP4 inhibitors were measured using real time quantitative PCR (RT-qPCR), as previously described [47]. Briefly, overnight cultures of *S. aureus* strains USA300, USA300Δ*pbp4*, CRB, and CRB∆*pbp4* were grown overnight in MH media, diluted (1:100 dilution) in fresh media and sub-cultured to a final OD_600nm_ of 0.2 at 37 °C with aeration. Cells were treated with 0 or 25 µM of the indicated compound for 3 h at 37 °C, which did not impact bacterial growth at these conditions. Cells were washed with PBS buffer, and collected via centrifugation at 1250× *g* for 10 min. Total bacterial RNA was isolated from cell pellets using Qiagen RNeasy kits, following the manufacturer’s guidelines for prokaryotic RNA isolation (Qiagen, Germantown, MD, USA). A total of 2 µg RNA was DNase treated and repurified using RNeasy kits. To measure *pbp4* expression, a total of 400 ng of total bacterial RNA was used as a template for qRT-PCR using *pbp4* primers (forward, 5′-GGAATCCAGCGTCTATGACTAAA-3′; reverse, 5′-GTCTCCTGCACCCATGATAAC-3′) and a total of 4 ng of total bacterial RNA was used as a template for primers specific for 16S rRNA (forward, 5′-ACGGTCTTGCTGTCACTTATAG-3′; reverse, 5′-CACTGGTGTTCCTCCATATCTC-3′). Bacterial RNA was amplified and measured using PerfeCTa SYBR Green FastMix and qScript cDNA SuperMix kits following the manufacturer’s recommendations (QuantaBio, Beverly, MA, USA). All samples were conducted in triplicate and normalized to 16S rRNA, averaged, and compared to untreated exponential phase wild type; RNA isolated from CRB∆*pbp4* was used as a negative control.

### 4.7. Triton X-100 Susceptibility Studies

The effect of putative PBP4 inhibitors on bacterial growth in media supplemented with Triton X-100 was measured, as previously described [25]. Briefly, an overnight culture of *S. aureus* strains USA300 was diluted to an OD_600nm_ of 0.03 in either fresh MH media, MH supplemented with 0.1% triton X-100 or MH supplemented with 12.5 µM of putative PBP4 inhibitor; DMSO treated cells served as a negative control, whereas strain USA300∆*pbp4* served as a positive control for triton susceptibility. Cell suspensions were incubated at 37 °C with aeration and cell density (OD_600nm_) was recorded hourly for a total of 7–8 h.

### 4.8. PBP4 BOCILLIN FL Binding Competition Assays

The ability of putative PBP4 inhibitors to impact the active site of recombinant PBP4 was assessed using two previously established BOCILLIN-FL Penicillin (Penicillin V conjugated to BODIPY fluorescent dye) binding assays. First, fluorescence polarization assays were performed as previously described, but with minor modifications [35]. Briefly, the indicated amount of the putative PBP4 inhibitor or control antibiotic was incubated with 1 µM recombinant *B. subtilis* PBP4 or *S. aureus* PBP2a in potassium phosphate buffer (40 mM K_2_HPO_4_ and 10 mM KH_2_PO_4_) for 1 h in black 96-well microtiter plates (Corning Life Sciences, Corning, NY, USA). Following incubation, 1 µM (final concentration) BOCILLIN FL penicillin was added, and the mixture was incubated for 30 min. BOCILLIN FL fluorescence polarization was measured at (495 nm excitation and 545 nm emission) in a SpectraMax M5 multimode plate reader (Molecular Devices, San Jose, CA, USA). All assays were repeated at least 8 times, averaged, and values falling outside of 2 standard deviations were considered outliers. The resulting data were plotted as percentage BOCILLIN FL binding inhibition = (T/U) × 100; T = FL polarization of ((PBP4+ BOCILLIN FL) − (PBP4+ BOCILLIN FL + inhibitor compound)), U = FL polarization of ((PBP4 + BOCILLIN FL) − (BOCILLIN FL)). Second, gel electrophoresis and fluorescence detection was used to measure BOCILLIN FL labeling of PBP4 treated with the known PBP4 inhibitor cefoxitin (positive control), kanamycin (negative control) or 9314848 (test compound), as previously described but with minor modifications [48,49]. Briefly, 1 µM of recombinant PBP4 was mixed with 0, 50, 500, 2000, or 5000 µM 9314848 or control antibiotic in 50 mM phosphate buffer (pH 7.4) and incubated at 37 °C for 30 min. Next 40 μM of BOCILLIN FL was added to the reaction mixture and incubated for an additional 30 min at 37 °C. Reactions were stopped by adding 5× SDS-PAGE gel loading dye, heated at 100 °C for 3 min and separated by electrophoresis in 12% SDS-PAGE (2 h, 120 Volts). BOCILLIN FL labeled PBP4 was imaged at λ = 365 nm using a BioRad Gel Doc EZ Imaging system and the gel was then subsequently stained with Bio-safe Coomassie Brilliant Blue stain (BioRad, Hercules, CA, USA) to measure the amount of protein present in each reaction condition and analyzed using NIH-ImageJ software. The percent BOCILLIN FL binding was calculated by first normalizing the protein present in each reaction condition to that of untreated control and subsequently applying each reaction’s protein normalization factor to each reaction’s fluorescent signal. The percentage BOCILLIN FL binding inhibition was then calculated as the normalized band intensity of antibiotic/compound treated sample as compared to the control sample (PBP4 + BOCILLIN FL in the absence of compound).

## Figures and Tables

**Figure 1 antibiotics-11-01351-f001:**
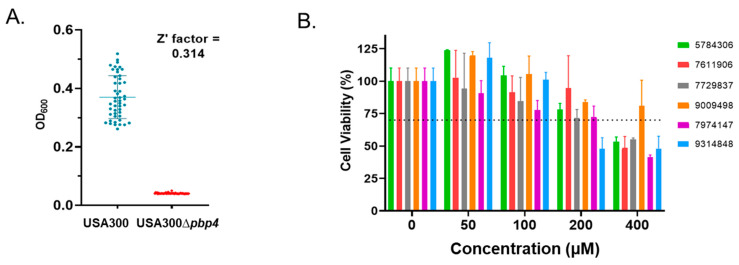
*S. aureus* PBP4 screening strategy and mammalian cell cytotoxicity of prioritized hits. Panel (**A**). Growth of strains USA300 and USA300∆*pbp4* in media supplemented with 2 µg/mL ciprofloxacin (48 replicates each). Panel (**B**). HepG2 cytotoxicity measures of 0, 50, 100, 200, and 400 µM of prioritized screening hits; standard deviation shown.

**Figure 2 antibiotics-11-01351-f002:**
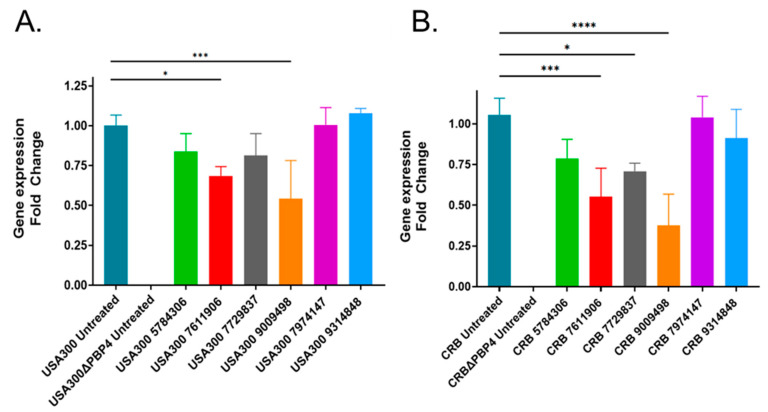
Effects of prioritized hits on *S. aureus* PBP4 expression. qRT-PCR measures of *S. aureus* strains USA300 (Panel (**A**)) and CRB (Panel (**B**)) *pbp4* mRNA titers following treatment with 25 µM treatment of prioritized hits. Corresponding results for *pbp4*-null strains shown; *t*-test (paired) *p* < 0.05 (*), *p* < 0.0005 (***), *p* < 0.0001 (****).

**Figure 3 antibiotics-11-01351-f003:**
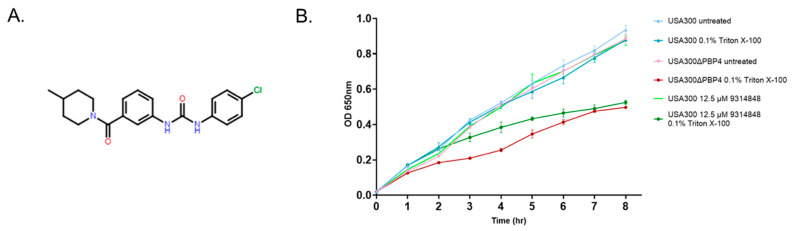
Characterization of highest priority putative *S. aureus* PBP4 inhibitor**.** Panel (**A**). Chemical structure of ChemBridge compound 9314848. Panel (**B**). Growth measures of *S. aureus* strains USA300, USA300∆*pbp4*, and 9314848 treated cells in the absence and presence of 0.1% of Triton-X100.

**Figure 4 antibiotics-11-01351-f004:**
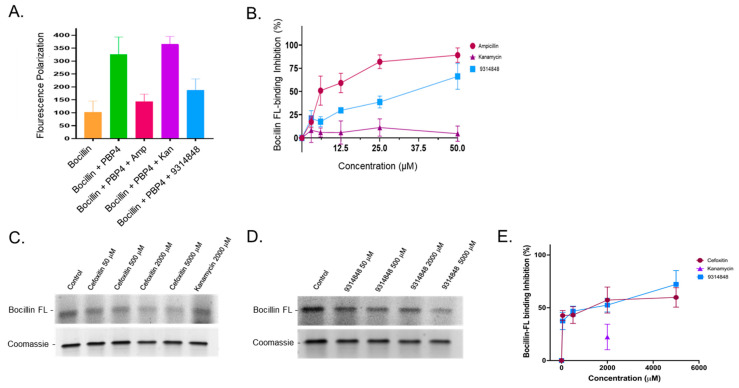
PBP4 binding assays. Panel (**A**). Fluorescence polarization of BOCILLIN FL (alone) and in the presence of *Bacillus subtilis* PBP4 following mock treatment or treatment with 50 µM of either ampicillin, kanamycin, or 9314848. Panel (**B**). Percent PBP4 BOCILLIN FL binding inhibition following pre-treatment with 0, 3.1, 6.2, 12.5, 25.0, or 50.0 µM ampicillin, kanamycin, or 9314848. Panel (**C**). *Top*—gel electrophoresis and fluorescent detection of BOCILLIN FL labeled PBP4 following treatment with 0 (control), 50, 500, 2000, or 5000 µM cefoxitin or 2000 µM kanamycin; *Bottom*—same gel Coomassie blue stained to control for loading. Panel (**D**). *Top*—gel electrophoresis and fluorescent detection of BOCILLIN FL labeled PBP4 following treatment with 0 (control), 50, 500, 2000, or 5000 µM 9314848; *Bottom*—same gel Coomassie blue stained. Panel (**E**). Graphed are the average gel-based BOCILLIN FL inhibition measures from 3 independent experiments; standard deviations shown.

**Figure 5 antibiotics-11-01351-f005:**
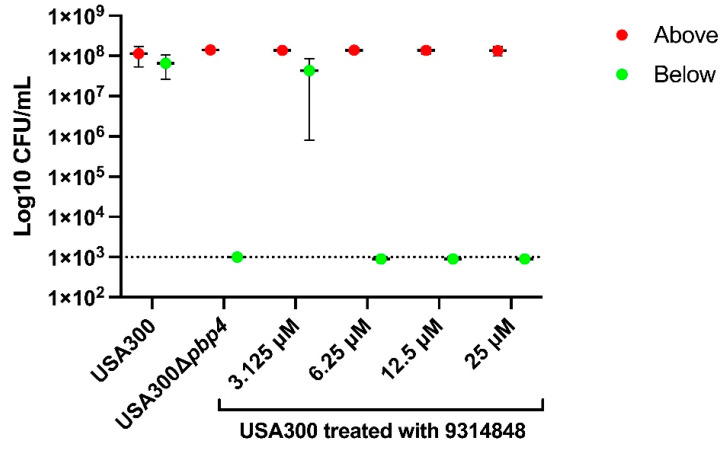
µSiM-CA assays**.** Enumeration of wild type USA300, USA300∆*pbp4*, and wild type cells treated with 0, 3.1, 6.2, 12.5, 25.0, or 50.0 µM 9314848 recovered.

**Table 1 antibiotics-11-01351-t001:** Minimum inhibitory concentration testing (µg mL^−1^).

Strain	NAF	MER	BPR	CIP	LEV	TOB	MIN	MUP	COL
USA300	8	4	0.5	4	1	1	0.5	0.125	128
USA300∆*pbp4*	4	2	0.25	1	0.5	1	0.5	0.125	128
USA300∆*pbp4* pPBP4	8	4	0.5	2	1	1	0.5	0.125	128
COLnex	0.25	1	0.25	0.25	0.25	0.06	0.5	0.03	128
CRB	>128	64	>128	0.5	0.5	0.06	0.5	0.03	128
CRB∆*pbp4*	0.25	1	0.25	0.25	0.25	0.06	0.5	0.03	128

NAF (nafcillin); BPR (ceftobiprole); MER (meropenem); CIP (ciprofloxacin); LEV (levofloxacin); TOB (tobramycin); MIN (minocycline); MUP (mupirocin); COL (colistin).

**Table 2 antibiotics-11-01351-t002:** Antimicrobial effects of ceftobiprole and putative PBP4 inhibitors.

Compound	CRB					CRB∆*pbp4*				
	Alone		Combination			Alone		Combination		
	BPR ^1^	Cmpd ^2^	BPR ^1^	Cmpd ^2^	Fold ^3^	BPR ^1^	Cmpd ^2^	BPR ^1^	Cmpd ^2^	Fold ^3^
5784306	128	100	4	12.5	32	0.25	400	0.25	12.5	0
7611906	128	400	16	25	8	0.25	400	0.25	25	0
7729837	128	200	64	50	2	0.25	400	0.25	50	0
9009498	128	100	8	25	16	0.25	100	0.06	25	4
7974147	128	400	8	25	16	0.25	400	0.25	25	0
9314848	128	400	8	25	16	0.25	400	0.125	25	2

^1^ BPR (ceftobiprole) µg/mL; ^2^ Cmpd (Putative PBP4 inhibitor) µM; ^3^ Fold-decrease in ceftobiprole resistance when in combination with the indicated putative PBP4 inhibitor.

**Table 3 antibiotics-11-01351-t003:** Bacterial Strains and plasmids.

Strain	Relevant Characteristics	Source
USA300	Community-associated MRSA	[42]
USA300∆*pbp4*	USA300 *pbp4* deletion	[32]
USA300∆*pbp4* pPBP4	USA300∆*pbp4* harboring pCN40-*pbp4*	[32]
COL	Hospital-associated MRSA	[43]
COLn	Tet^S^ COL derivative	[33]
COLnex	COLn ∆SCCmec	[33]
CRB	COLnex PBP4 over-expresser derivative	[19]
CRB∆*pbp4*	CRB *pbp4* deletion	[24]

Tet^S^—tetracycline susceptible.

## Data Availability

Raw data is available on request.

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
