# Peer review of "Identification of Staphylococcus aureus Penicillin Binding Protein 4 (PBP4) Inhibitors"

_antibiotics, 2022, doi:10.3390/antibiotics11101351_

Round 1
Reviewer 1 Report
The manuscript "Identification of Staphylococcus aureus Penicillin Binding Pro-2 tein 4 (PBP4) inhibitors" by Young et al has depth and will contribute to the scientific community. the authors are required to do additional experiments. It would be really important to show the Spectrum of activity of the selected compound of author's choice. The authors can select a list of compounds (maybe 5 or so) from their final screen and show the activity in question against other bacteria or fungi strains that have related PBP1 or 2 or 3 or PBP4 genes (70-90% match in the genome sequence of any bacteria such as Escherichia coli, Pseudomonas aeruginosa, Mycobacterium smegmatis, Proteus vulgaris, or fungi strains e.g. Enterococcus faecalis, etc). This will suggest that most of the inhibitors are specific to S. aureus and may or may not be effective against diverse bacterial species. It may be possible that the inhibitory effect of the selected compound is conserved in S. aureus and functional orthologs are not found in other bacteria or fungi.
Line 170: Can the author define what is Fractional inhibitory concentration? FIC versus MIC testing differences?
Line 230: the cell viability max of 70% seems high. Please define this cut-off value selection!
Figure 4 C, D, E: How the binding inhibition (%) in E was normalized using Bocillin-FL to Coomasie? needs to explain in methods.
Line 335: define cana-licular colonization. why it is required to test?
Line 398-400 please rewrite the sentence not clear. The approach taken was inspired by earlier work that revealed that the PBP4-centric antibiotic, cefoxitin, is capable of restoring β-lactam susceptibility toward community-acquired MRSA [Memmi et al.].
Line 578: put comma as appropriate (Molecular Devices,San Jose, CA).
please check the typos and errors throughout the manuscript.
Line 582: what gel-based assays - please be specific.
Line 592-595: How author normalised these values? The binding of Bocillin-FL to PBP4 was analyzed using (Line 592) NIH-ImageJ software, and percent Bocillin-FL binding inhibition was calculated as the (Line 593) percentage of band intensity of antibiotic/compound treated sample as compared to con- (Line 594) trol sample (PBP4 + Bocillin-FL in the absence of compound).
Author Response
Reviewer: The manuscript "Identification of Staphylococcus aureus Penicillin Binding Pro-2 tein 4 (PBP4) inhibitors" by Young et al has depth and will contribute to the scientific community. the authors are required to do additional experiments. It would be really important to show the Spectrum of activity of the selected compound of author's choice. The authors can select a list of compounds (maybe 5 or so) from their final screen and show the activity in question against other bacteria or fungi strains that have related PBP1 or 2 or 3 or PBP4 genes (70-90% match in the genome sequence of any bacteria such as Escherichia coli, Pseudomonas aeruginosa, Mycobacterium smegmatis, Proteus vulgaris, or fungi strains e.g. Enterococcus faecalis, etc). This will suggest that most of the inhibitors are specific to S. aureus and may or may not be effective against diverse bacterial species.
Response- We thank the reviewer for this very thoughtful suggestion, which frankly, is an excellent extension of the current work that we had not previously considered. In response, we have now tested the antibiotic potentiating effects of our front runner S. aureus PBP4 inhibitor compound, 9314848, toward a second Gram-positive and a Gram-negative pathogen of immediate healthcare concern, Enterococcus faecalis and Pseudomonas aeruginosa. Our results (below) show that, as expected, the β-lactam potentiating effect of the PBP4 inhibitor is S. aureus specific. We have discussed this new data within the Discussion section of the revised manuscript (Lines 427-429).
|
|
Pseudomonas aeruginosa PAO1 |
|
Enterococcus faecalis V583 |
|
|||||||||||||
|
|
Alone |
|
Combination |
|
Alone |
|
Combination |
||||||||||
|
Compound |
Antibiotic1 |
93148482 |
|
Antibiotic 1 |
93148482 |
Fold3 |
|
Antibiotic1 |
93148482 |
|
Antibiotic1 |
93148482 |
Fold3 |
|
|||
|
Ceftobiprole |
4 |
>400 |
|
4 |
>400 |
0 |
|
0.5 |
>400 |
|
0.5 |
>400 |
0 |
|
|||
|
Nafcillin |
>128 |
>400 |
|
>128 |
>400 |
0 |
|
8 |
>400 |
|
8 |
>400 |
0 |
|
|||
|
1Antibiotic µg ml-1; 29314848 (Putative PBP4 inhibitor) µM; 3Fold-decrease in antibiotic resistance when in combination with the indicated putative PBP4 inhibitor (9314848). |
|||||||||||||||||
Reviewer: Line 170: Can the author define what is Fractional inhibitory concentration? FIC versus MIC testing differences?
Response- Fractional Inhibitory Concentration testing is a standard procedure to measure the antimicrobial performance of compound combinations as being synergistic, additive, indifferent or antagonistic, and the corresponding experimental approach has been described in detail (Materials Lines 511-527 of original manuscript). Nonetheless, we recognize that the reviewer is correct that the text could be made clearer. Thus, we have revised line 170 to read “Standard fractional inhibitory concentration testing to measure compound interactions…”. Whereas MIC is minimum inhibitory concentration.
Reviewer: Line 230: the cell viability max of 70% seems high. Please define this cut-off value selection!
Response- Cell viability cut-off values have historically varied within the literature. We have applied and followed the International Organization for Standardization guidelines, which define 70% as a cut-off value for toxicity. This is indicated on line 231-232 of the revised manuscript.
Reviewer: Figure 4 C, D, E: How the binding inhibition (%) in E was normalized using Bocillin-FL to Coomasie? needs to explain in methods.
Response- We would note that previous publications using gel based Bocillin-FL binding inhibition measures do not include loading standards (Coomassie results). We strongly believe that they should be included within manuscripts, such as we have done here. We thank the reviewer very much for pointing out that as initially written, application of the Coomassie results were qualitative as opposed to quantitative. In response, we have edited lines 591-600 of the revised methods text to clarify that “Bocillin-FL labeled PBP4 was imaged at λ = 365 nm using a BioRad Gel Doc EZ Imaging system and the gel was then subsequently stained with Bio-safe Coomassie Brilliant Blue stain (BioRad, Hercules, CA, USA) to measure the amount of protein present in each reaction condition and analyzed using NIH-ImageJ software. The percent Bocillin-FL binding was calculated by first normalizing the protein present in each reaction condition to that of untreated control and subsequently applying each reaction’s protein normalization factor to each reaction’s fluorescent signal. The percentage Bocillin-FL binding inhibition was then calculated as the normalized band intensity of antibiotic/compound treated sample as compared to control sample (PBP4 + Bocillin-FL in the absence of compound).
Reviewer: Line 335: define cana-licular colonization. why it is required to test?
Response- We are somewhat puzzled by this query or are perhaps missing the point of the comment. As elaborated within the introduction (Lines 81-97), S. aureus canalicular colonization is mediated by PBP4 and is required for reoccurring osteomyelitis. The overarching goal of this study was to identify small molecule inhibitors of PBP4 which could serve as progenitors of a new class of antimicrobials that interfere with the organism’s ability to colonize canalicular networks and consequently chronic osteomyelitis. Thus, we have tested whether the PBP4 inhibitors that we have discovered reduce entry into canalicular networks using previously established in vitro methods. They (PBP4 inhibitors we have discovered) do indeed seem to inhibit this process very effectively; we are quite excited.
Reviewer: Line 398-400 please rewrite the sentence not clear. The approach taken was inspired by earlier work that revealed that the PBP4-centric antibiotic, cefoxitin, is capable of restoring β-lactam susceptibility toward community-acquired MRSA [Memmi et al.].
Response- we respectfully disagree with the referee and feel that it is very important to recognize: 1. The previous work of others, 2. Specifically acknowledge that their work inspired us to use antibiotic susceptibility as a screening method for identifying putative PBP4 inhibitors.
Reviewer: Line 578: put comma as appropriate (Molecular Devices,San Jose, CA).
Response- revised
Reviewer: please check the typos and errors throughout the manuscript.
Response- revised
Reviewer: Line 582: what gel-based assays - please be specific.
Response- Respectfully, that is how the assays are referred in the literature and we have specified the exact method in Materials and methods (Lines 583-599; beginning with “Second, gel-based assays were used to…”).
Reviewer: Line 592-595: How author normalised these values? The binding of Bocillin-FL to PBP4 was analyzed using (Line 592) NIH-ImageJ software, and percent Bocillin-FL binding inhibition was calculated as the (Line 593) percentage of band intensity of antibiotic/compound treated sample as compared to con- (Line 594) trol sample (PBP4 + Bocillin-FL in the absence of compound).
Response- As noted above, we have edited lines 591-600 of the revised text to clarify how the data were normalized.
Reviewer 2 Report
Young et al. present the discovery of a PBP4 inhibitor in a high-throughput compound screening. The manuscript describes all the necessary steps for validating compound 9314848 as an inhibitor of PBP4. On top of that, the manuscripts describes the compounds effect on limiting S. aureus membrane transmigration and cortical bone osteocyte lancuno-canalicular network colonization.
This manuscript is significant for the scientific community and by its publication other members of the community can pick the thread at the point where Young et al. conclude the effectiveness of the compound 9314848, giving future researchers the chance to produce protein-ligand crystal structures and to perform studies with this compound in combination with S. aureus murine infection models.
Author Response
Reviewer: This manuscript is significant for the scientific community and by its publication other members of the community can pick the thread at the point where Young et al. conclude the effectiveness of the compound 9314848, giving future researchers the chance to produce protein-ligand crystal structures and to perform studies with this compound in combination with S. aureus murine infection models.
Response: The reviewer seemed to appreciate the work and had no suggestions for improving the manuscript.
Reviewer 3 Report
Comments to the authors
Based on the previous discovery that PBP4 is critical in MRSA and might play some roles in OLCN invasion by S. aureus, Young et. al. sought to develop a high-throughput model to screen potential inhibitors of PBP4 from a chemical library of 30000 compounds. Using yeast and mammalian cell cytotoxicity assays, transcriptional analysis, binding analysis and μSiM-CA model, the authors found one lead candidate that was low cytotoxicity, possibly inhibited PBP4 activity, and could prevent MRSA USA300 strain propagation in μSiM-CA canaliculi model. This study is interesting and practically important to the drug development for the rise of MRSA infections. However, the manuscript can be improved further if the authors could address my comments below.
Major:
1. From reading the paper, it seems to me that using CRB strain is a much more logic and more specific model than using USA300 and CIP to screen PBP4 inhibitors. What is the rationale for using the latter screen approach? Is there any mechanism why PBP4 inhibitors could enhance the killing by CIP? If the inhibitor increases cell permeability (which could also unspecifically promote many drugs’ entry), but why it increased bacterial susceptibility to only CIP drugs? Does this happen by chance? The authors might need to elaborate more on this issue.
2. Related to the first question, why USA300 strain which has PBP4 is not a good screen model to PBP4 inhibitors using beta-lactam drugs?
3. Although Z’ factor test is often used in high-throughput screens, could the authors explain in Figure 1A caption what each date point was? Why USA300 strain had big variations? Is Z factor of 0.314 a good indicator?
4. Figure 2: Did those compounds inhibit bacterial growth in the absence of drugs? Bacterial growth might significantly affect the transcription of PBP4?
5. How is B. subtilis PBP4 related to S. aureus PBP4? I know they are both gram-positive bacteria, but what is the amino acid similarity between the two proteins? The authors might need to justify why the results obtained by the former bacteria will also be true in the latter bacteria.
6. It would be great for the authors to add some explanation about the mechanism by which compound 9314848 interact with PBP4? Does the compound competitively or non-competitively inhibit PBP4? What is the Kd or EC50 for the binding? A way to test it would be using a constant Bocilline-Fl with different concentration of the compound added to the reactions, following by measuring the decrease of fluorescence.
7. Figure 5, why the change of CFU was so stiff from 3.125 μM and 6.25 μM treatments? Similarly, In Figure 4E, big change from concentration 0 to 50 μM? Is it a reversible binding between the dye and PBP4? Between the hit compound and PBP4?
Minor:
Line 119(and below) “ Table 2” This the first Table mentioned in the manuscript, Isn’t it Table 1 ?
Lines 306: use “to” verify the impact ….
Line 329: panel E or F?
Author Response
Reviewer: This study is interesting and practically important to the drug development for the rise of MRSA infections. However, the manuscript can be improved further if the authors could address my comments below.
Response: We are glad that the reviewer appreciated the overarching study and are thankful for the thoughtful suggestions provided for improving the manuscript's presentation (below).
Reviewer: From reading the paper, it seems to me that using CRB strain is a much more logic and more specific model than using USA300 and CIP to screen PBP4 inhibitors. What is the rationale for using the latter screen approach?
Response- The reviewer is 100% correct. In fact, we initially hoped to use CRB for screening for all the reasons the reviewer indicated. The reason we did not was simply because we were unsure of whether it would be possible to get the strain; ultimately it took over 1 year and a laborious path to get the strain, through no one’s fault. It turns out that Dr. Chambers (created CRB) retired and was slow to respond to our strain requests. We subsequently learned that the strain was passed to Dr. Chamber’s former trainee, Dr. Chatterjee, who had moved to another institution and who ultimately provided us with the strain. For that reason, we started the project with USA300 but switched to CRB when it became available and specifically state in Lines 185-186 of the revised manuscript “using … strains that became available during our study (generous gift from Dr. S. Chatterjee; University of Maryland)."
Reviewer: Is there any mechanism why PBP4 inhibitors could enhance the killing by CIP? If the inhibitor increases cell permeability (which could also unspecifically promote many drugs’ entry), but why it increased bacterial susceptibility to only CIP drugs? Does this happen by chance? The authors might need to elaborate more on this issue.
Response: The reviewer astutely brings up a point that we have also struggled to understand, which is why would a PBP4 mutant display increased susceptibility to ciprofloxacin in the USA300 background? We do not know, nor have we formally evaluated, as that is not the goal of the immediate work. To the reviewer’s point, in lines of 403-411 of the revised manuscript we state “Using a MRSA USA300 strain as a model screening organism we found that indeed PBP4 modulates the organism’s β-lactam resistance phenotype, but that the protein was associated with greater resistance to the fluoroquinolone, ciprofloxacin, which was an unexpected finding that, to our knowledge, has not been previously reported. Nonetheless, others have found that PBP4 mediates Erwinia sp. fluoroquinolone resistance (William Johnson, personal communication). Presumably, the previously observed decrease in peptidoglycan crosslinking of PBP4 deficient cells facilitates the antibiotic’s cellular entry, although this was not formally evaluated and was not a generalizable phenotype for other antibiotic classes, such as mupirocin, that target cytoplasmic enzymes.”
Reviewer: Related to the first question, why USA300 strain which has PBP4 is not a good screen model to PBP4 inhibitors using beta-lactam drugs?
Response- Simply because the PBP4 had a greater magnitude of effect (4X) vs. ciprofloxacin in comparison t beta-lactams (2X). Thus, ciprofloxacin has a greater dynamic range. We've modified lines 132-133 of the revised manuscript to more clearly articulate why we selected ciprofloxacin as opposed to beta-lactams for our primary screen.
Reviewer: Although Z’ factor test is often used in high-throughput screens, could the authors explain in Figure 1A caption what each date point was? Why USA300 strain had big variations? Is Z factor of 0.314 a good indicator?
Response- We thank the reviewer for pointing out Z’ factor testing data points need to be clarified. In Figure 1A legend we have now indicated that each data point is a replicate (48 per strain). A Z' factor score of 0.31 is considered good, but not excellent (Z' factor score of 0.5).
Reviewer: Figure 2: Did those compounds inhibit bacterial growth in the absence of drugs? Bacterial growth might significantly affect the transcription of PBP4?
Response- We thank the reviewer for pointing out that we failed to indicate that treatment was only for 3 hrs at concentrations that were 4-16x lower than needed to affect growth at 16 hrs. We have edited line 547 of the revised manuscript to indicate these assay conditions had no measurable impact on bacterial growth.
Reviewer: How is B. subtilisPBP4 related to S. aureus PBP4? I know they are both gram-positive bacteria, but what is the amino acid similarity between the two proteins? The authors might need to justify why the results obtained by the former bacteria will also be true in the latter bacteria.
Response- Simply put, we would have preferred to use S. aureus protein. In fact we have cloned the gene for expression purposes but we have been unable to successfully purify it despite numerous attempts, presumably due to its transmembrane domains. For that reason, we were relegated to using the only commercially available S. aureus PBP4 ortholog, which happens to be from B. subtilis. We are currently contracting with another commercial company to purify all of the S. aureus penicillin-binding proteins; they are also having issues.
Reviewer: It would be great for the authors to add some explanation about the mechanism by which compound 9314848 interact with PBP4? Does the compound competitively or non-competitively inhibit PBP4? What is the Kd or EC50 for the binding? A way to test it would be using a constant Bocilline-Fl with different concentration of the compound added to the reactions, following by measuring the decrease of fluorescence.
Response: We have now provided the IC50 for inhibition of the Bocillin-FL binding (Line 304) and now edited the text (lines 352-353 of the revised text) to indicate the EC50 for uSIM-CA assays is between 3.125 and 6.25 uM. We would note that, while the compound provides impressive PBP4 inhibitory activities in the assays used here, it is merely a hit in a screen. It is refreshing that the reviewer identified exactly what we are currently doing, which is to identify more potent analogs of compound 9314848 as a means to more robustly define the chemical series' mechanism of action (using both biochemical and genetic approaches). To that end, we have now created 30 compound analogs and generated more potent derivatives that are the focus on a follow-on manuscript, complete with detailed mechanism of action data and defined binding site. We feel that work is beyond the scope of the current study (and is a stand alone manuscript).
Reviewer: Figure 5, why the change of CFU was so stiff from 3.125 μM and 6.25 μM treatments? Similarly, In Figure 4E, big change from concentration 0 to 50 μM? Is it a reversible binding between the dye and PBP4? Between the hit compound and PBP4?
Response: Through the use of 30 analogs (described above; some of which are more potent PBP4 inhibitors, whereas others are weaker PBP4 inhibitors) we have used structural predictors that indicate the chemical series may be a competitive inhibitor, we are currently designing a third generation of analogs to test that hypothesis. Currently that work, which is a massive undertaking (30 analogs synthesized and processed through all the assays described here and structural biology), is the focus of a follow-on manuscript.
Minor:
Reviewer: Line 119(and below) “ Table 2” This the first Table mentioned in the manuscript, Isn’t it Table 1 ?
Response: Based on our understanding of the author guidelines, the Table numbering is correct; Table 1 contains the strains used in this study. Table 2 indicates the MIC values of those strains.
Reviewer: Lines 306: use “to” verify the impact ….
Response: Revised.
Reviewer: Line 329: panel E or F?
Response: Revised (Panel E).
Round 2
Reviewer 1 Report
The authors have resolved most queries but the Following Comments were not resolved to satisfy the reviewer-
Line 398-400 please rewrite the sentence not clear. The approach taken was inspired by earlier work
Please check the typos and errors throughout the manuscript. Specify with Line Numbers where all typos and errors are corrected.
Line 582: what gel-based assays - please be specific.
Author Response
We thank the reviewer for the thoughtful comments for improving the manuscript's presentation. Below we've listed the reviewer's three comments and how they have been addressed:
Reviewer: Line 398-400 please rewrite the sentence not clear. The approach taken was inspired by earlier work
Response: Line 401 of the revised manuscript- the sentence reading “The approach taken was inspired by earlier work that revealed that the PBP4-centric antibiotic, cefoxitin, is capable of restoring β-lactam susceptibility toward community acquired MRSA [25]. By extension…” has been changed to “To that end, it has been previously shown that cefoxitin, a β-lactam that binds PBP4 with high affinity, reverses the PBP4 mediated oxacillin resistance phenotype of community acquired MRSA strains [25]. By extension…”
Reviewer: Please check the typos and errors throughout the manuscript. Specify with Line Numbers where all typos and errors are corrected.
Response: The following typos/errors were corrected:
Line 12: (c.m.) changed to (C.M.)
Line 85: lancuno changed to lacuno
Table 2 legend: naficillin changed to nafcillin
Line 191: (Colnex) changed to (COLnex)
Line 198: Colnex changed to COLnex
Line 230: hepatocarcinoma changed to hepatocellular carcinoma
Line 314: ceftoxitin changed to cefoxitin
Line 321: ceftoxitin changed to cefoxitin
Line 385: lancuno changed to lacuno
Line 418: “expressor” changed to expresser
Line 428: “ceftobioprole” changed to ceftobiprole
Line 429 “ceftobioprole” changed to ceftobiprole
Table 1. “expressor” changed to expresser
Line 519: “hrs” changed to hr
Line 540: “hrs” changed to hr
Line 548: “hrs” changed to hr
Line 559: “PerfeCta” changed to PerfeCTa
Line 559: “Fastmix” changed to FastMix
Lines 572,580, 581, 585, 586, 587, 588, 589, and 592: “Bocillin-FL” changed to BOCILLIN FL
Reviewer: Line 582: what gel-based assays - please be specific.
Response: Line 588 of the revised manuscript: The sentence reading “Second, gel-based assays were used to measure BOCILLIN-FL binding to PBP4, as previously described but with minor modifications [50, 51]. Briefly…” has been changed to “Second, gel-electrophoresis and fluorescence detection was used to measure BOCILLIN FL labeling of PBP4 in the absence or presence of the known PBP4 inhibitor cefoxitin (positive control), kanamycin (negative control), or 9314848 (test compound), as previously described but with minor modifications [50, 51]. Briefly,…”